# The Role of the Design of Public Squares and Vegetation Composition on Human Thermal Comfort in Different Seasons a Quantitative Assessment

Priscila Weruska Stark da Silva [1],*, Denise Duarte [2] and Stephan Pauleit [1],*

[1] Chair for Strategic Landscape Planning and Management, School of Life Sciences, Technical University of Munich, Emil-Ramann-Str. 6, 85354 Freising, Germany

[2] School of Architecture and Urbanism, University of Sao Paulo, Rua do Lago, 876 Cidade Universitaria, São Paulo 05508-080, SP, Brazil

\* Correspondence: priscila.stark@tum.de (P.W.S.d.S.); pauleit@tum.de (S.P.)

**Abstract:** Increasingly, public open spaces are gaining importance for human well-being in dense, urban areas. In inner city locations, squares can provide easy access to greenery and thus encourage social encounters. Microclimatic conditions influence the squares' attractiveness. However, knowledge is still limited on the impact of different layouts of squares, particularly the impact of the vegetation composition on the human thermal comfort across the seasons in temperate climates. Therefore, our research aims to discern how human thermal comfort is affected by the different elements existing in different open areas of Munich, Germany. For this purpose, five different squares were analyzed on five typical days to create an overview of how human thermal comfort is affected by the layout and vegetation composition during the year. The study areas were selected in view of their size, pavement type, and the number of trees. Micrometeorological simulations were performed using the ENVI-met V 4.6 model to identify how different aspects affect the physiological equivalent temperature (PET) on typical Munich days. The urban morphology was observed to be the greatest factor affecting PET in all the cases studied. Of microclimate variables, the surface temperature was relevant only on warm days. Long-wave radiation, on the other hand, positively affected the PET on cold days. The results suggested that urban morphology has a high impact on the human thermal comfort in urban squares. The results obtained showed that it is necessary to consider diverse vegetation arrangements combined with urban morphology characteristics to optimize human thermal comfort under a range of climatic conditions.

**Keywords:** green infrastructure; urban climate; outdoor thermal comfort; microclimate modeling; ENVI-met

## 1. Introduction

In modern societies, people spend about 90% of their lives indoors where thermal conditions for most of their activities can be controlled [1]. However, with city densification, private areas have become smaller, making access to a network of green open spaces an essential factor in improving the quality of life in cities. Compact city designs are often justified as a way to save land from exploitation; however, greener designs also need to be considered during city planning processes. Urban areas that lack green spaces are difficult to change later [2].

Several studies emphasize the benefits of green spaces on human mental health, reducing physiological stress and the risk of mortality [3,4]. Green spaces are also associated with reduced mortality from cardiovascular causes, possibly through mediators, such as lower air pollution, decreased stress, increased human thermal comfort, physical activity, social contact, and recovery [5]. Additionally, the COVID-19 pandemic highlighted the physical and mental benefits of recreational access to urban green spaces when travel

restrictions, home offices, and the demand for social distance demonstrated the need for suitable urban spaces that are easily accessible [6].

Urban green spaces have also been studied regarding their benefits in lowering risks from heat stress in human health [7]. This highlights the importance of vegetation coverage to human thermal comfort [8–10].

Human thermal comfort is the influence of the thermal environment on the human body. Among the available thermal indexes, the physiological equivalent temperature (PET), developed by Höppe and Mayer [11], combines the variable mean radiant temperature (MRT), air moisture, air velocity, and the heat balance of the human body [12].

It is not difficult to find studies that consider urban parks as their research focus, considering their benefits for human thermal comfort [8,13,14]; however, less research has been dedicated to squares, which are easier to build and can be more evenly distributed in the inner city due to their smaller size and consequently lower costs for construction and maintenance.

Most studies have focused on the green designs of public squares and their microclimatic influences during a hot summer's day to compare the thermal comfort values [10,15,16], others have explored the climate change adaptation potential in landscape architectural designs [17], whilst others have analyzed the diverse thermal performances of several urban squares with varying morphological and green characteristics [18,19]. Only a few studies, however, have balanced both summer and winter conditions [20–25].

There is still a need for studies to explore the influence of different vegetation types on the human thermal comfort of squares across the seasons. Among the available studies, Irmak et al. [26] analyzed human thermal comfort during summer and winter in Erzurum, Turkey, using RayMan software [27] to model PET [11]. Further, Huang et al. [28] performed their analysis using recorded images complemented by RayMan simulations to understand the square users' preferences in summer and winter. Empirical research combining field measurements and questionnaires was developed in different seasons and climates to confirm the human thermal adaptation concept and the relation between the thermal environment and square use [21,29,30]. Xu et al. [22] observed the benefits of deciduous trees to the Universal Thermal Climate Index UTCI [31] under winter conditions, while Kariminia et al. [15] explored the influence of the built environment's geometry on the thermal comfort, supporting the results observed in our study. These findings suggest that urban morphology is the factor with the most significant impact on human thermal comfort in urban squares, as also observed by Zhang et al. [32] in their spatiotemporal study of the thermal environment in Singapore.

Therefore, this study aims to help fill the knowledge gap regarding how human thermal comfort is affected by the design of existing urban squares, in different seasons of a temperate climate in order to determine how human thermal comfort is affected by the pavement and vegetation types. This study analyzes the potential impact of different urban greenery types and the presence of pavements in distinguished sizes of urban squares on the human thermal comfort of typical urban squares. Our main focus is the interactions between the different elements that compound the existing squares. Five urban squares in densely built-up neighborhoods of Munich, Germany, were selected for this study. We analyzed the differences in size and shape, the number and arrangement of trees, the pavement type and boundaries, the microclimate, and the thermal comfort under different climatic conditions. Using the ENVI-met model, we simulated the parameters used to characterize the selected urban squares on a microscale to measure the impact of each parameter on the human thermal comfort of urban squares under summer and winter climatic conditions.

## 2. Materials and Methods

### 2.1. Study Area

Munich is the third-largest city in Germany with more than one and a half million inhabitants. The city is located in the south of the country (48°8′23″ N, 11°34′28″ E, elevation

519 m a.s.l.), with an annual average air temperature of 9.7 °C and an average precipitation of 944 mm, following the reference period 1981–2010 [33]. The climate classification is the Cfb category of the Köppen-Geiger. The city's climate includes hot summers, the absence of dry seasons, and higher precipitation rates during the summer. Munich's city ventilation and urban temperature distribution are influenced by a thermal wind system, also known as alpine pumping [34].

Munich covers an area of 310 km$^2$ and has 4.223 ha of public green space (circa 13.6% of the city area) [35].

Public spaces in the city center of Munich are often intensively used and, because of their diverse structures, they have very different effects on people's individual and thermal well-being [36]. In Project 100 Places:M (2020), Munich's urban squares and their main structural attributes were recorded. From this study, five squares were selected for this study that clearly differ in size, pavement type, and in the number of trees (Figure 1).

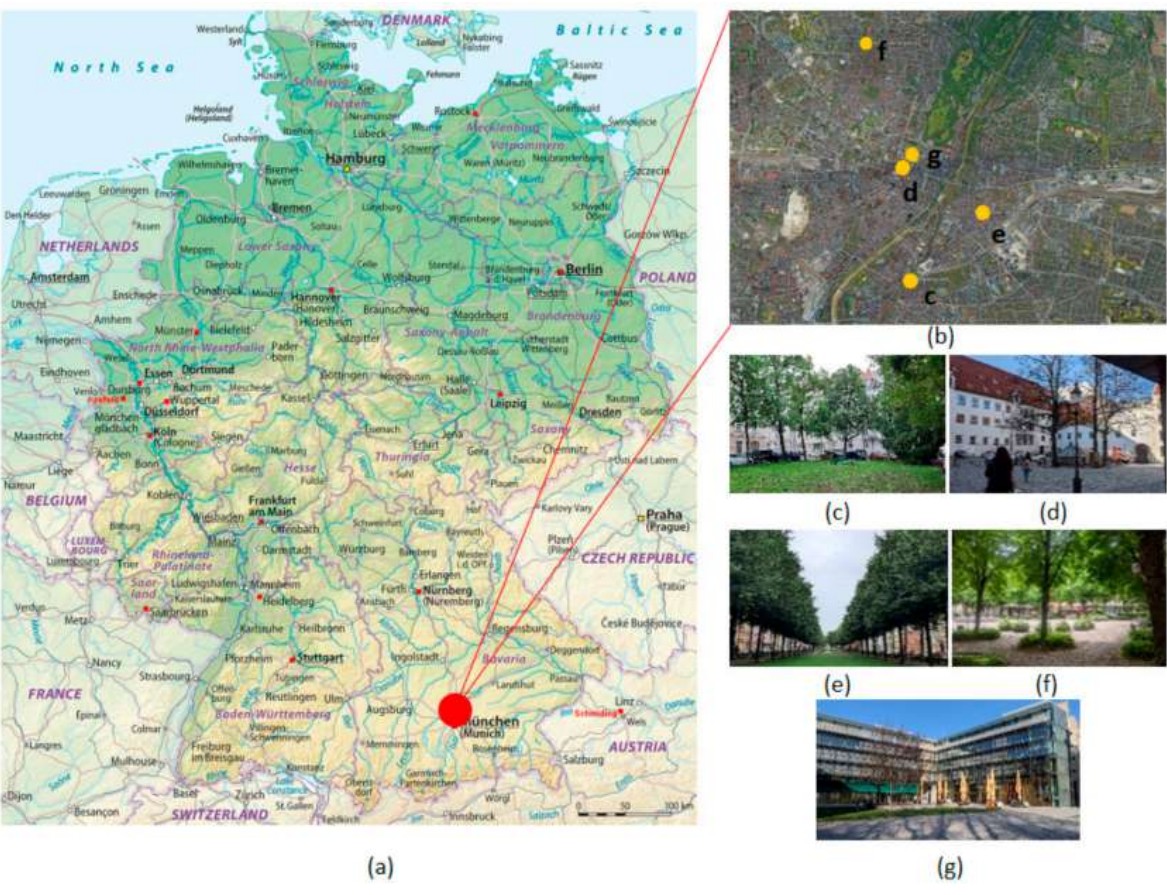

**Figure 1.** Location of Munich in Germany (**a**) and Munich's selected squares (**b**); Alpenplatz (**c**); Alter Hof (**d**); Bordeauxplatz (**e**); Hohenzollernplatz (**f**); Marstallplatz (**g**). Source: Priscila W. Stark da Silva (with basic geographical data provided by www.google.com and www.flickr.com accessed on 5 February 2023).

Alpenplatz and Alter Hof represent small squares differing in the number of trees, type of pavement, and boundaries. While Alpenplatz has a high vegetation cover, Alter Hof is highly paved, with only three trees, and is surrounded by buildings on all sides. Bordeauxplatz, Hohenzollernplatz, and Marstallplatz represent the large squares in this study. Bordeauxplatz and Hohenzollernplatz have a high number of trees, contrasting with Marstallplatz, which has few trees considering its area. Regarding the pavement type, while Bordeauxplatz has 20% of impervious surface, Hohenzollernplatz and Marstallplatz are highly paved.

Table 1 shows the parameters considered to select the squares studied in this research work.

**Table 1.** Variables considered for square selection (Source: 100Places:M).

| Square | Area (m²) | Impervious Percentage | Number of Trees | Boundaries |
| --- | --- | --- | --- | --- |
| Alter Hof | 3546 | 100% | 3 | Buildings |
| Alpenplatz | 3317 | 20% | 30 | Streets |
| Bordeauxplatz | 14,018 | 30% | 84 | Streets |
| Hohenzollernplatz | 10,150 | 89% | 90 | Buildings and streets |
| Marstallplatz | 9517 | 91% | 18 | Buildings |

*2.2. Methods*

This study uses micrometeorological modelling on a micro scale to investigate the outdoor human thermal comfort for five real urban squares in different seasons.

The following flowchart (Figure 2) describes the procedures of the experiment, from the problem analysis that defined our approach followed by the literature review, the variables considered for the selection of the squares and the days to analyze, how these data were combined to develop the microclimatic simulation, ending with a comparative analysis and conclusions. A detailed approach is presented in the following subtopics.

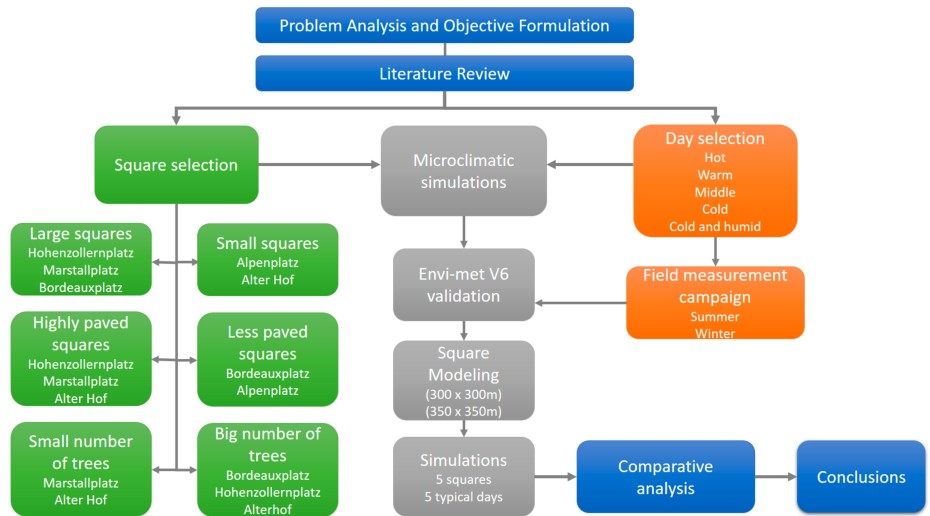

**Figure 2.** Flowchart of the methodological approach.

2.2.1. Day Selection

Munich's average air temperatures vary between 0.9 °C in January and 19.6 °C in July, and the annual average precipitation is 938.9 mm, following the reference period 1991 to 2020 of the German Weather Service (DWD) [33]. In order to carry out the intended meteorological simulations in this study, it was necessary to identify days representing extreme and typical weather situations throughout the year. For this reason, DWD data obtained from station 3379—Munich City (WST), year 2020 were investigated. Rainy days were ignored as they would have led to unreliable results in our microclimatic modeling approach. Therefore, longer periods of weather stability were analyzed. Among them, at least four days without rain were considered. Then the days with the most regular air temperature and humidity curves were grouped according to their similar data (Figure 3). From them, we defined the five typical days, called cold, cold and humid, mild, warm, and hot.

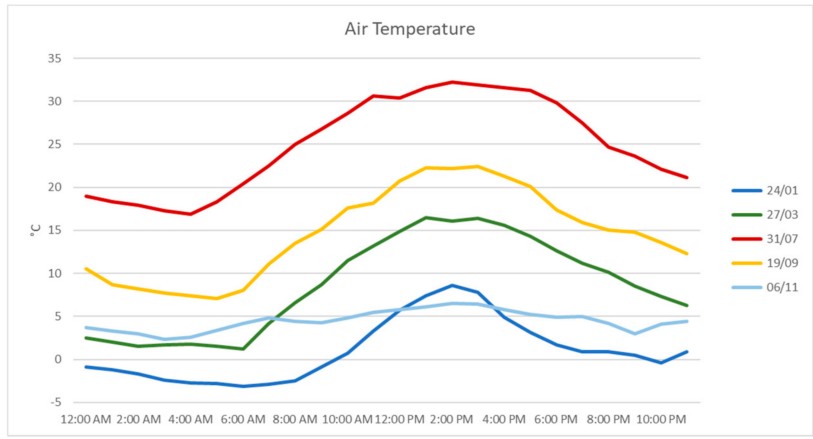

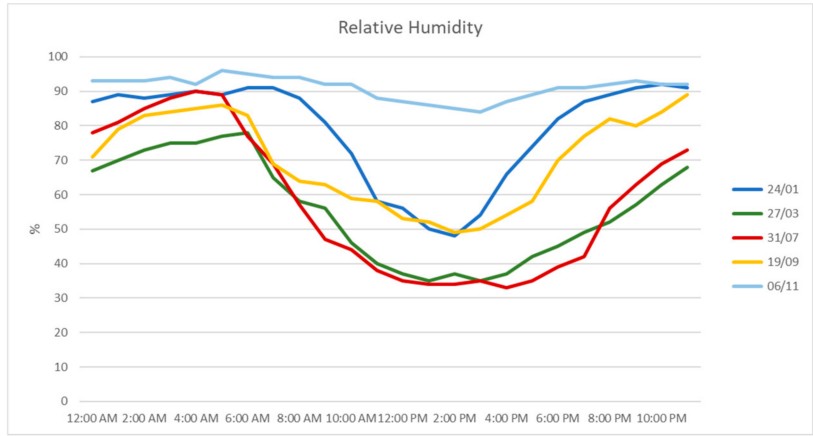

**Figure 3.** Typical air temperature and relative humidity of selected days.

### 2.2.2. Micrometeorological Simulation of the Squares Using the ENVI-Met Model

All simulations were performed with the three-dimensional microscale model ENVI-met [37], version 4.4.6. The ENVI-met is a high-resolution microclimate model that simulates the interactions among soil, vegetation, and atmosphere on a microscale. Based on principles of fluid mechanics and thermodynamics, it is able to calculate three-dimensional (3D) variables: turbulence, air temperature and humidity, radiation fluxes, and the dispersion of pollutants [37]. The square models of Alter Hof, Alpenplatz, Bordeaux-platz, and Hohenzollernplatz were sized 300 × 300 m. Because of the size of Marstallplatz square, its model was sized 350 × 350 m. For higher accuracy, the horizontal resolution was 2 × 2 m, without nesting grid. The vertical resolution was 3 m with an equidistant grid. Building heights and dimensions were derived from the GIS-Data provided by the City of Munich. Tree species were provided by the Project 100 Places:M (2020), LAD values were used according to the predefined species in the ENVI-met, while the floor materials, asphalt and granite, were identified through site visits (Figure 4).

The simulations started at 2 am for a total time of 48 h. The first 22 h of the analysis were excluded to overcome the initial transient conditions. The simulation results were analyzed for four different hours (2 am, 8 am, 12 pm, and 4 pm). The results for human thermal comfort were extracted at a height of 1.5 m at a pedestrian level. The meteorological data required for the ENVI-met simulation were obtained from the DWD's weather station, City-Station ID 3379 (Table 2).

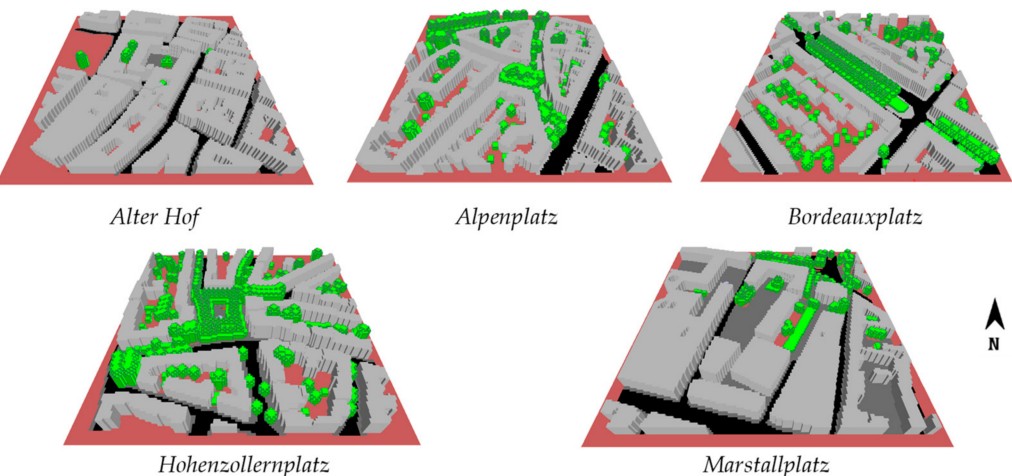

**Figure 4.** 3D graphic representation of the studied square models.

**Table 2.** ENVI-met model setup and meteorological input data.

| Day Classification | Cold | Mild | Hot | Warm | Cold and Humid |
|---|---|---|---|---|---|
| Start of simulation | 23 January 2020 | 26 March 2020 | 30 July 2020 | 18 September 2020 | 5 November 2020 |
| Duration of simulations | 48 h | | | | |
| Min/Max Ta | −3.9/8.6 °C | −2.0/16.5 °C | 17.6/33.7 °C | 7.1/22.4 °C | 2.3/11.3 °C |
| Min/Max RH | 48/95% | 35/80% | 33/91% | 49/91% | 84/97% |
| Daily sum of solar incoming radiation | 287 J/cm² | 1770 J/cm² | 2676 J/cm² | 1786 J/m² | 636 J/cm² |
| Model grid size/resolution | | | Alter Hof, Alpenplatz, Bordeauxplatz, Hohenzollernplatz 150 × 150 × 25 (x,y,z)/Vertical equidistant grid Marstallplatz 172 × 172 × 25 (x,y,z)/Vertical equidistant grid | | |
| Building material | | | Default wall—moderate insulation | | |
| Soil material | | | Sandy clay loam, granite, asphalt with gravel | | |
| Relative soil humidity | | | Upper layer: 70%, Middle and deep layers: 75% | | |
| Lateral boundary conditions | | | Full forcing | | |

For the vegetation composition, we used standard tree species that represent the trees found in the squares, such as deciduous *Tilia cordata* and *Acer platanoides*, two of the most common tree species in the squares and streets in Munich. For grass, we selected the default option: grass, 25 cm average density.

We used the full forcing option to allow for observation of the impact of several variables, such as wind speed on the PET values. The Forcing Manager option of the ENVI-met model permits the creation of specific full forcing files of your own. Full Forcing allows the forcing of all the predefined profiles in the main model, including air temperature, humidity, wind speed and direction, radiation or cloud cover, and precipitation.

### 2.2.3. Model Validation

As noted in the previous studies [38–41], preliminary sensitivity tests are recommended to verify the model's response to the input parameters. Using data measured on 16 August 2020 and 12 December 2021, the model calibration was performed to compare the existing conditions in Hohenzollernplatz and Bordeauxplatz via point-to-point verification of air temperature and relative humidity in cold and hot conditions.

In order to test the validity of the model under the most extreme weather conditions, field measurement campaigns were conducted in August 2020 and December 2021 with two similar self-built weather stations equipped with HOBO and Ecomatik sensors and iButton Hygrochron Temperature/Humidity Logger. During the day, the mobile weather station sensors recorded the air temperature, relative humidity, wind speed (at 2 m height), black globe temperature (at 1.4 m height), and surface temperature with a 10 s sampling and 5 min recording interval. The iButton high-resolution temperature and humidity measurements

recorded data during the day and the night after the previous 24 h of calibration time (Figure 5).

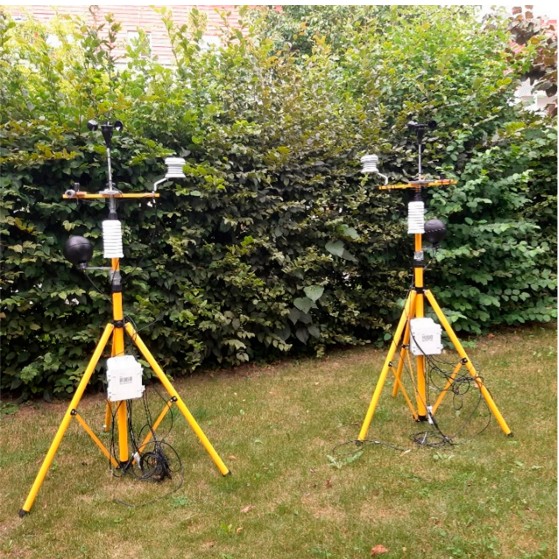

**Figure 5.** iButton sensor calibration.

The weather stations were placed at Hohenzollernplatz and Bordeauxplatz squares, which differ in shape and the type of pavement. In both locations, the stations were placed in open areas, as representative unshaded areas within the study site, surrounded by vegetation, and distant from streets and building facades (Figure 6). The measured summer period was representative of a hot day with maximum air temperature higher than 29 °C, no clouds, and wind speed below 2 m/s. The measured winter period was representative of a typical cold day, average Ta 0.6 °C and wind speed around 1 m/s (Figure 7).

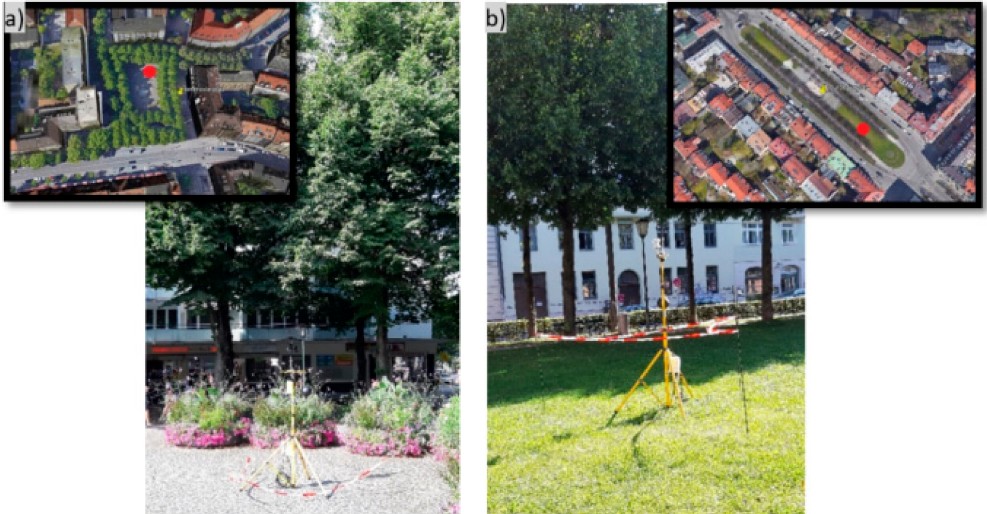

**Figure 6.** Summer field measurement. Hohenzollernplatz (**a**); Bordeauxplatz (**b**). Source: Priscila W. Stark da Silva (with basic geographical data provided by www.google.com accessed on 5 February 2023).

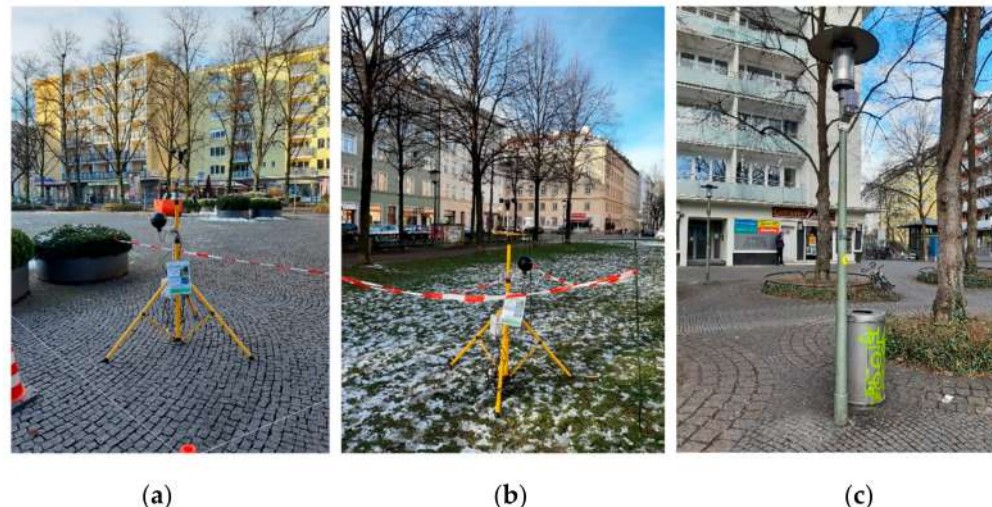

**Figure 7.** Winter field measurement. Hohenzollernplatz (**a**); Bordeauxplatz (**b**); iButton (**c**).

2.2.4. Statistical Analysis

In order to complement the simulation results analysis, a multivariate regression was used to determine how the variables: air temperature, wind speed, surface temperature, and relative humidity affected the human thermal comfort. These variables were analyzed to quantify how the squares' individual characteristics, i.e., the type of pavement, number of trees, and boundaries affected the PET values due to their impact on the micrometeorological variables during day and night conditions. The statistical analysis was developed using Microsoft Excel.

**3. Results**

For the standardization of the results and the consequent comparisons, all the simulations were performed using the same input data obtained from the DWD database for the defined days. The simulation results were then compared in order to understand how the different aspects of each square impacted the human thermal comfort. The human thermal comfort was analyzed according to the PET interpretation scale suggested by Matzarakis and Mayer [42], (Table 3).

**Table 3.** PET range interpretation according to Matzarakis and Mayer (1996).

| PET (°C) | <4 | 4~8 | 8~13 | 13~18 | 18~23 | 23~29 | 29~35 | 35~41 | >41 |
|---|---|---|---|---|---|---|---|---|---|
| Grade of physiological stress | Extreme cold stress | Strong cold stress | Moderate cold stress | Slight cold stress | No thermal stress | Slight heat stress | Moderate heat stress | Strong heat stress | Extreme heat stress |

*3.1. Hot Day*

On the hot day, in Alter Hof, the simulation results revealed intense thermal stress conditions (mean PET 43.8 °C) during the day for the critical period analyzed (12 pm and 4 pm). The simulation result also shows that at night the maximum PET value at 2 h was observed under the trees (16.7 °C). The warming was maintained due to the canopy that kept the longwave radiation trapped and, consequently, the PET was higher when compared to the open areas. At 12 pm and 4 pm, the lowest PET values (34.9 °C) were observed in the shaded areas of the buildings, while the highest PET values were observed in the unshaded areas (62.3 °C). In contrast, at Alpenplatz, the simulation results revealed mild thermal stress conditions (mean PET 26.6 °C) during the critical period analyzed (12 pm and 4 pm). On the hot day, the benefits of the trees and grass are clearly observed in the reduction in the PET, with a difference of 23.6 °C between the shaded and the unshaded areas where the maximum PET values reached 46.0 °C at 12 pm (Figure 8).

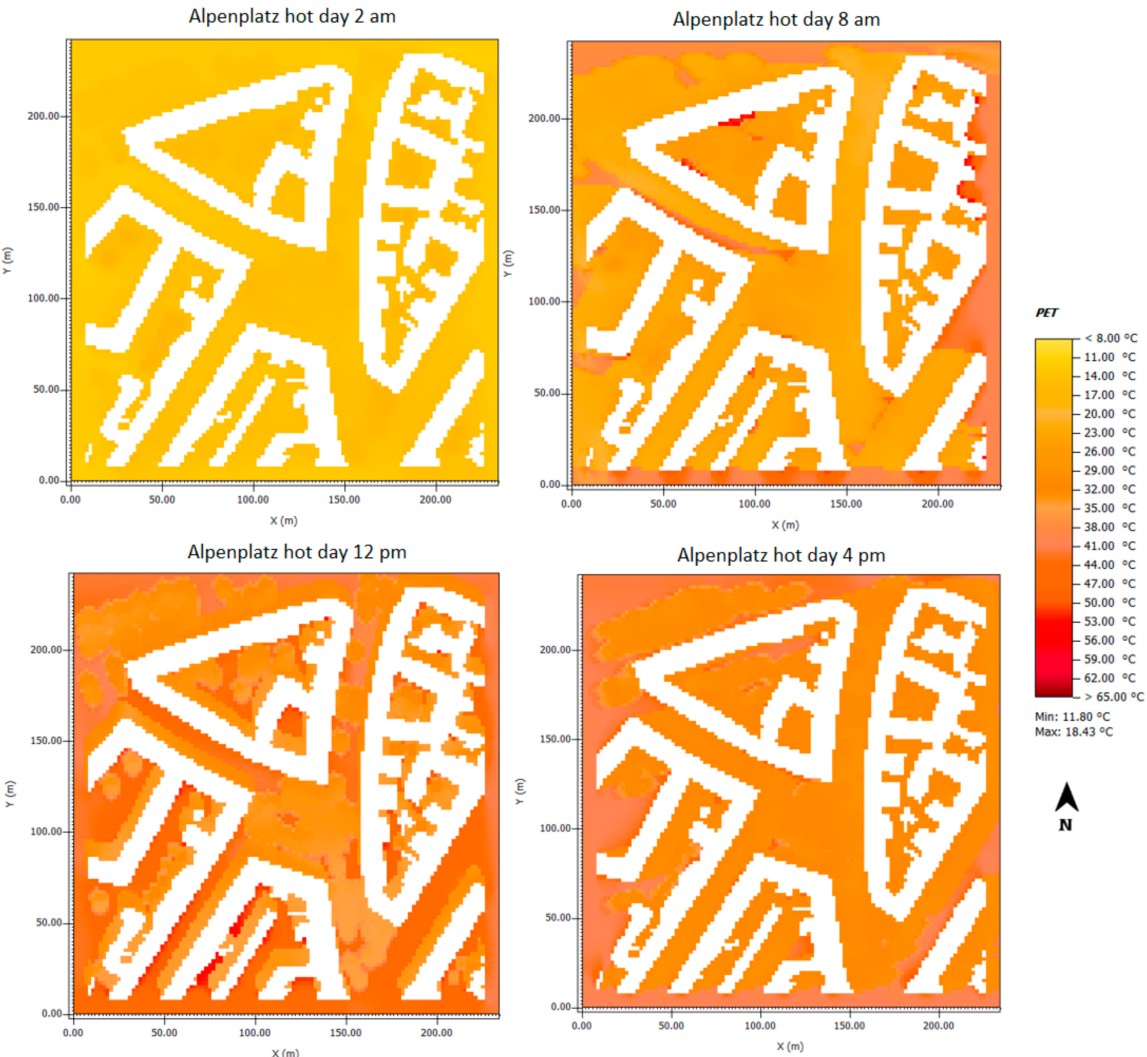

**Figure 8.** Alpenplatz's PET variation during the hot day (2 am, 8 am, 12 pm, 4 pm).

On the hot day, despite the high amount of vegetation at Bordeauxplatz, the simulation results revealed strong thermal stress conditions (mean PET 40.3 °C) during the critical period analyzed (12 pm and 4 pm). As also seen at Alter Hof, the highest PET value (20.5 °C) was observed under the trees at night. The double effect of the shading from the trees and from the buildings reduced the PET values in almost the entire canyon at 4 pm (mean PET 38.0 °C). At Hohenzollernplatz, even with the high percentage of a paved area, the shadow effect was responsible throughout the hot day for a mild heat stress condition (mean PET 28.5 °C).

As defined by Watson and Johnson [43], the Sky View Factor (SFV) is the ratio of radiation received from the sky by a planar surface in relation to that received from the entire hemispheric radiating environment. At night, the lowest PET value (13.4 °C) was observed in the center of the square, highlighting the role of the SVF for the release of the longwave radiation at nighttime. For the same reason, the highest PET value (35.9 °C) was also observed at 8 am under the trees. The wind turbulence area presented in Marstallplatz was reflected in the lowest PET values on the hot day (12.8 °C) (Figure 9) and on the cold day (−1.02 °C) (Figure 10). Due to Marstallplatz's highly paved area and the small number of trees, on the hot day the simulation results revealed extreme heat stress conditions (mean PET 61.1 °C) between 12 pm and 4 pm. On this day, higher PET values were also observed in the northern area between the trees and the building at 8 am (39.3 °C), while at 4 pm,

lower PET values were observed in the overlapping shaded area by the building and the trees (34.3 °C).

**Figure 9.** Marstallplatz's PET variation during the hot day (2 am, 8 am, 12 pm, 4 pm).

Figure 11 shows the air temperature at the five squares on the hot day. The regression analyses showed that the wind speed had the highest-impact variable on the PETs during the critical times (12 pm and 4 pm) at Alter Hof (>77%), Bordeauxplatz (>88%), Hohenzollernplatz (>86%), and Marstallplatz (>79%), while the surface temperature was the highest-impact variable at Alpenplatz (>62%).

At night, the simulation result shows that all the squares are in slight cold stress conditions (<15 °C), with Marstallplatz showing the lowest average PET (14.6 °C). The result highlights the effect of the high SVF in reducing the PET at night. The regression analyses revealed that wind speed was the variable with the highest impact on the PET at night at Alpenplatz (99%), Hohenzollernplatz (89%), and Marstallplatz (97%). The effect of the paving on the surface temperature at night was also observed. The lower albedo of the granite was responsible for the higher surface temperatures observed during the day (mean ST 32.5 °C) and the longwave radiation released at night. The Hohenzollernplatz, a highly paved and vegetated square, reinforces the effect of the canopy on heat retention by increasing the mean PET at night (15.2 °C) because of its reduced sky-view factor, despite the high influence of wind speed at night.

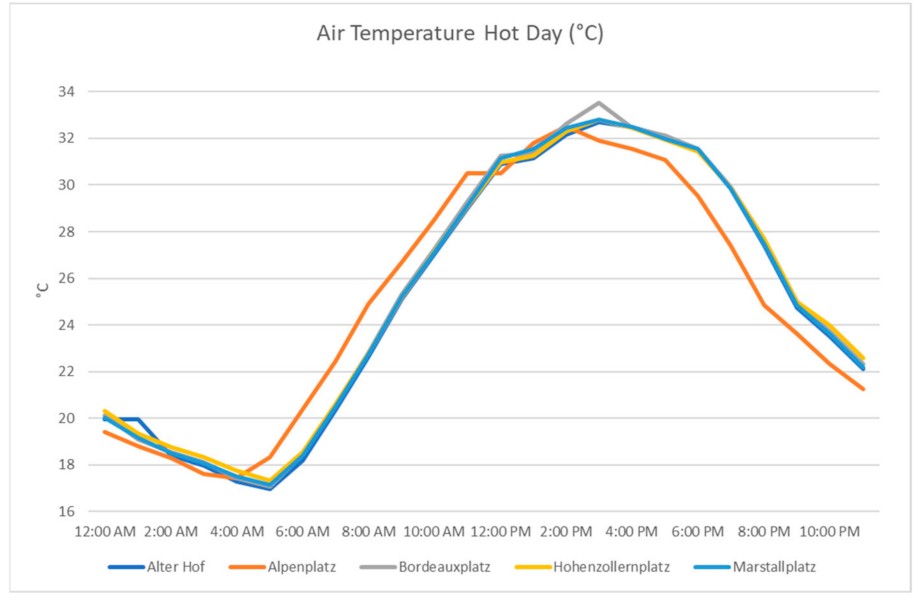

**Figure 10.** Marstallplatz's PET variation during the cold day (2 am, 8 am, 12 pm, 4 pm).

**Figure 11.** Simulated air temperature on the hot day.

During the critical hours (12 pm and 4 pm) all the squares, except Alpenplatz, are under extreme thermal stress (mean PET 41.2 °C). This is possibly because Alpenplatz presents a double shading effect from the trees and the buildings provided by the SVF. The lack of shading provided by the buildings influenced the PET values at Bordeauxplatz at critical times during the day. When compared to Hohenzollernplatz, a square with a similar number of trees, but highly paved, Bordeauxplatz shows higher PET values especially at 12 pm as can be observed in Figure 12.

PET hot day 12 pm

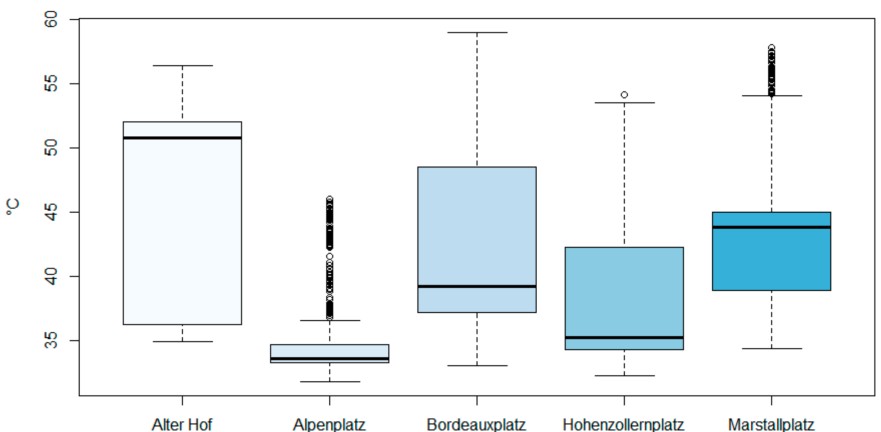

PET hot day 4 pm

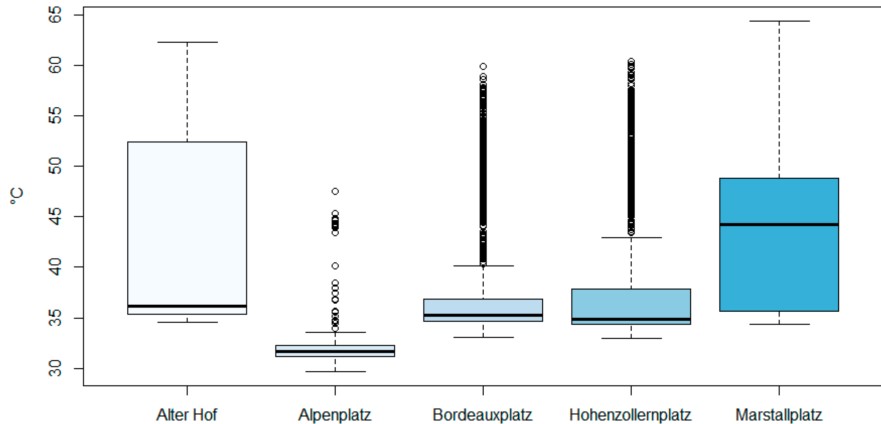

**Figure 12.** PET values for the hot day at critical hours (12 pm and 4 pm).

At Alpenplatz, the presence of grass and trees combined with the SVF all provide strong protection from shortwave radiation, while benefiting from the wind. All these factors keep Alpenplatz under moderate thermal stress (average PET 33.5 °C) during the critical hours of the day.

### 3.2. Warm Day

At night and at 8 am, all the squares are under moderate cold stress and due to the sky view factor, Alpenplatz has the highest PET at 2 am (Figure 13). While Alpenplatz, Bordeauxplatz, and Hohenzollernplatz have no thermal stress during the most critical hours (12 pm–4 pm). Alter Hof and Marstallplatz have significant areas under slight heat stress at 12 pm (PET 23~29 °C), highlighting the importance of tree shade and surface for human thermal comfort.

PET warm day 2 am

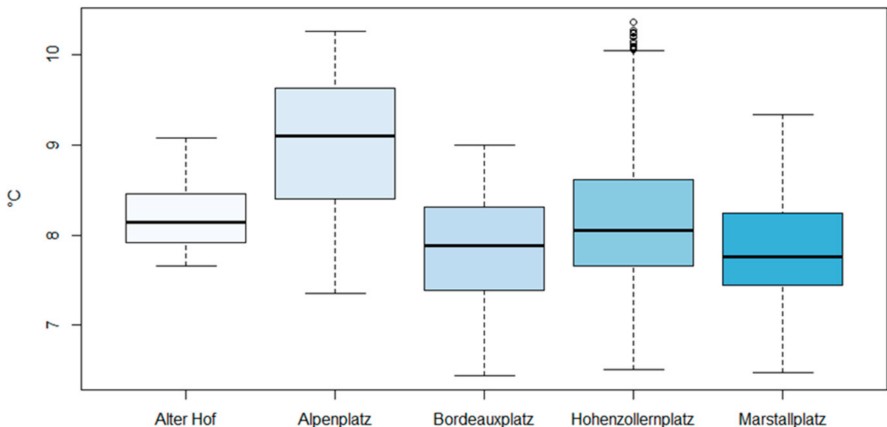

**Figure 13.** PET values for warm day at night (2 am).

At 4 pm, building shade is the main factor in reducing discomfort at Alter Hof and provides no thermal stress conditions (Figure 14). The building shade reduces the surface temperature and, according to the regression analysis, is the variable that has the strongest impact (80%) on PET at 4 pm.

PET warm day 4 pm

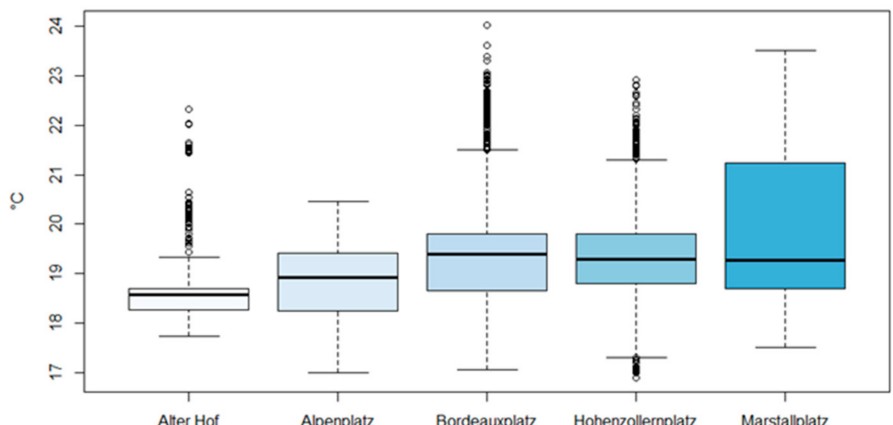

**Figure 14.** PET values for warm day 4 pm.

*3.3. Mild Day*

At night and at 8 am the simulation results show that all the squares are under moderate cold stress conditions (PET 8~13 °C), with Marstallplatz presenting the lowest average PET at 2 am and 8 am. During the day, the PET at Marstallplatz at 12 pm only shows an unusual performance, with the highest average value of 23.8 °C. In all the other cases, the squares have similar average values. The simulation also reveals that between 12 pm and 4 pm all the squares are in the comfort range (PET 18~23 °C). At 8 am the PET of the squares is under moderate to cold stress (PET 8~13 °C), with the surface temperature being the variable with the strongest influence on the PET values at Bordeauxplatz (95%), while at the other squares, the air temperature is the variable with the strongest influence (between 60 and 86%). At 12 pm, while the PET of the other squares is under a slight cold stress condition (mean PET 13~18 °C), Marstallplatz is under slight thermal stress (mean PET 23.0 °C). The maximum PET at Marstallplatz, 34.6 °C, is observed near the buildings, demonstrating the clear effect of longwave radiation. The simulated results show that at 4 pm all the squares are under moderate cold stress (PET 8~13 °C) (Figure 15)

and, according to the regression analysis, the air temperature is the variable that affects the PET the most at Alpenplatz (91%), Bordeauxplatz (96%), and Marstallplatz (81%). At Alter Hof, the PET is most affected by the surface temperature (80%), and at Hohenzollernplatz it is most affected by the wind speed (81%).

PET mild day 12 pm

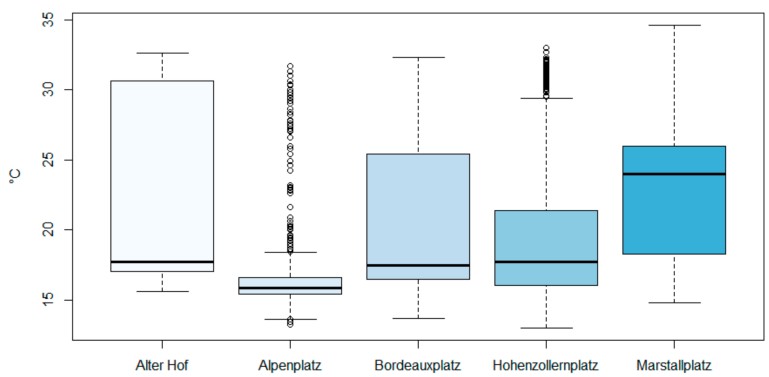

PET mild day 4 pm

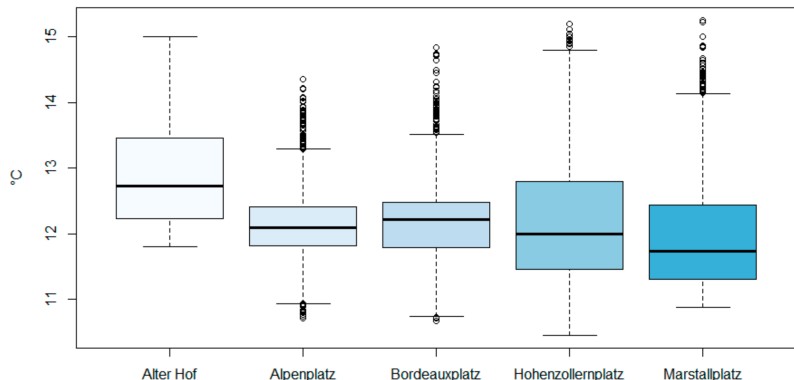

**Figure 15.** PET values for mild day 12 pm and 4 pm.

### 3.4. Cold Day

The observed PET at all the squares on the cold day ranges from strong cold stress at 12 pm to extreme cold stress in the other investigated hours. At Alter Hof, strong cold thermal stress conditions were observed (average PET 4.3 °C), while the highest PET values were observed close to the building walls (7.6 °C at 12 pm), confirming the effect of the longwave radiation and the wind protection provided by the Alter Hof design, a small and highly paved square surrounded by buildings on all sides. While Alpenplatz was under extreme cold thermal stress conditions (mean PET 2.7 °C), the minimum observed PET value was −1.0 °C at 8 am, caused by the lack of solar radiation at night. When compared to the Alter Hof results, the wind at Alpenplatz is not blocked, which decreases the PET values throughout the day.

At Bordeauxplatz, the simulation results revealed extreme cold thermal stress conditions (mean PET 2.3 °C) throughout the analyzed period. As observed at Alter Hof, on the cold day, the highest PET values were observed near the walls, confirming the effect of longwave radiation. At Hohenzollernplatz the simulation results revealed extreme cold stress conditions throughout the cold day (mean PET 2.1 °C), while Marstallplatz exhibited extreme thermal stress conditions (mean PET 3.3 °C). The square has a particular design, with a second open area located in the northwest of the site, which is part of the main access

of the Bavarian Academy of Sciences (Figure 16). The connecting location between the two open areas provides an area of wind turbulence that affects the PET values in this area.

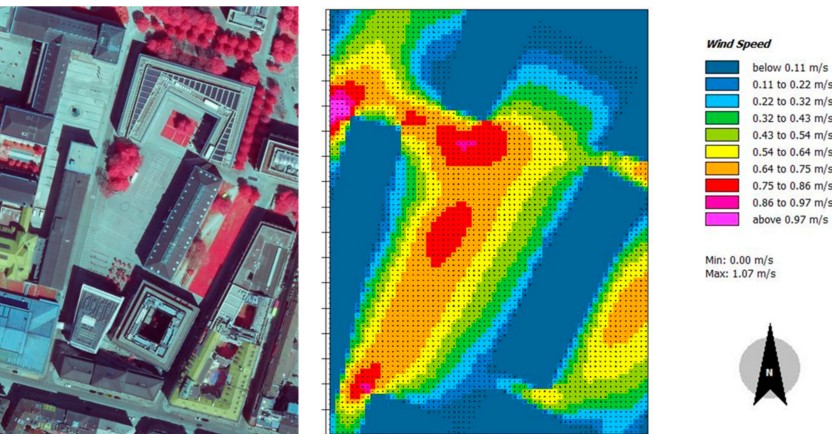

**Figure 16.** Aerial view and 8 am wind speed simulation of Marstallplatz (Source: Bayerische Vermessungsverwaltung).

The air temperature is the variable that most affects the PET on the cold day (at least 82%), and the second variable that most affects the PET is the wind speed (between 88% at 4 pm and 99% at 2 am). The importance of wind protection on cold days is observed in Alter Hof, which shows the highest values of PET during the whole cold period analyzed (Figure 17).

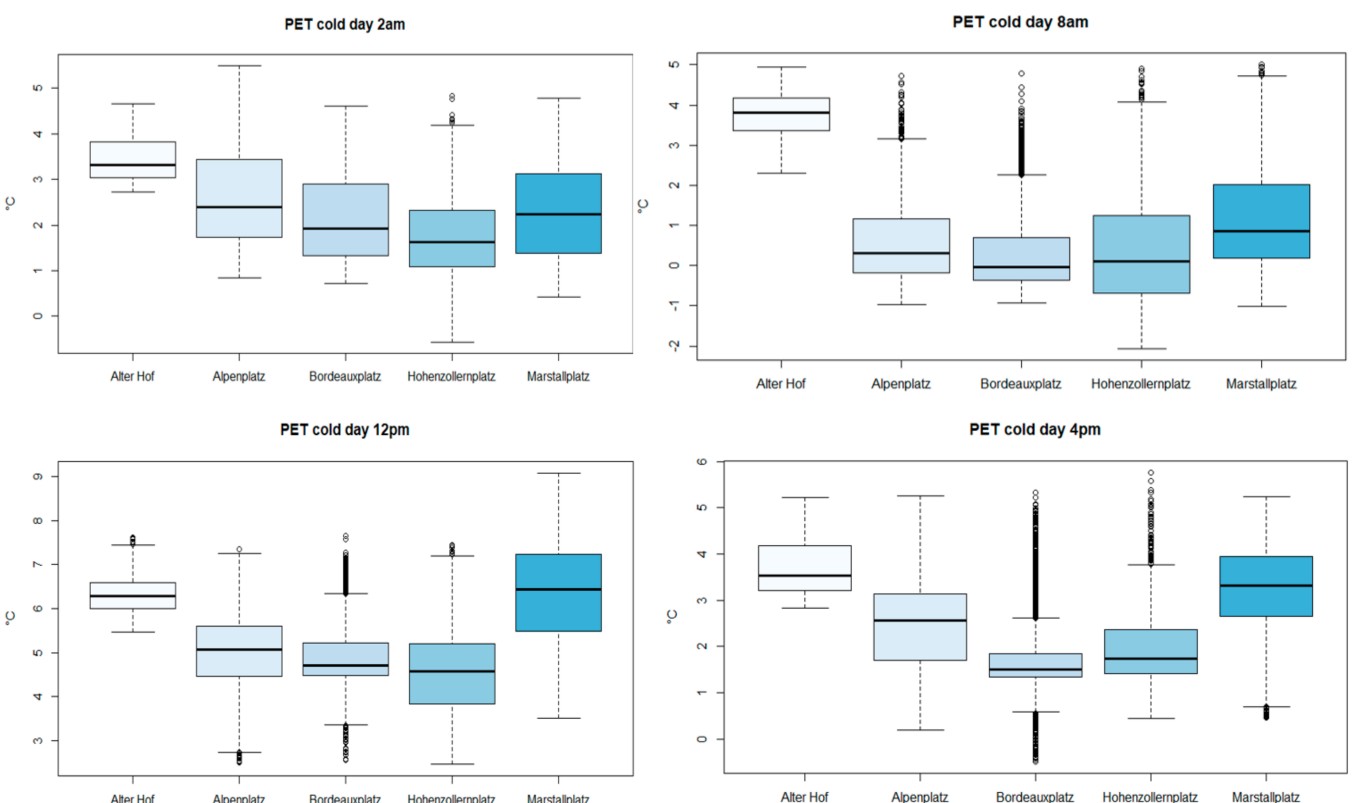

**Figure 17.** PET values for cold day.

### 3.5. Cold and Humid Day

Despite the low temperatures, the cold and humid day cannot be directly compared to the cold day. The input air temperature range of the simulation is distinct, ranging from −3.9 °C to 8.6 °C on the cold day and 2.3 °C to 11.3 °C on the cold and humid day. Since the input data are actual data it reflects on the distinct PET results on the cold and humid day. The PET at 2 am, 8 am, and 4 pm is under strong cold stress at most squares, with Marstallplatz showing extreme cold stress only at 2 am, with an average PET of 3.6 °C. At 12 am, all the squares are in the moderate to cold stress range, with the highest mean value observed at Marstallplatz (13.5 °C).

The wind speed is the variable that affects the PET the most at 2 am, 8 am, and 4 pm (lowest radiation times), according to the regression analysis, even with the low wind speed values observed in Figure 18.

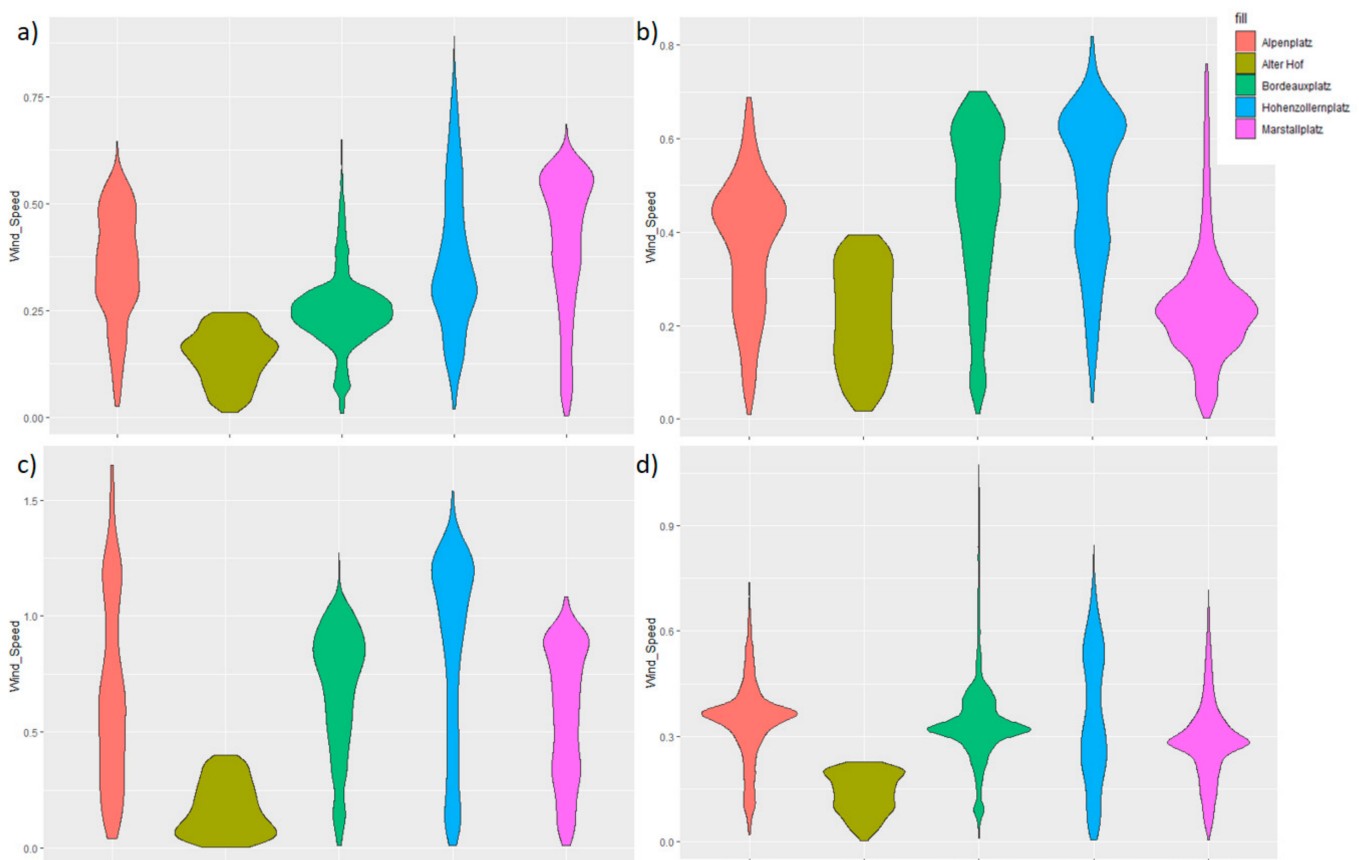

**Figure 18.** Wind speed comparison on the cold and humid day 2 am (**a**); 8 am (**b**); 12 pm (**c**); 4 pm (**d**).

The PET of all the squares is most affected by the air temperature at 12 pm (between 66 and 85%); at other times, the variables affect the PET differently depending on the square. Only at Marstallplatz is the air temperature the variable that most affects the PET at all times.

## 4. Discussion

By comparing the five different squares on five typical Munich days to create an annual overview it was possible to identify the different relevant characteristics depending on the season. Kariminia and Sh. Ahmad [15] in their study of Isfahan, Iran, which has a moderate and dry climate, observed the urban morphology effect in different climates due to the variation in the height to width ratio (H/W) of the squares. The urban morphology was also observed in our study as having an important impact on the PET since the thermal energy is strongly affected by the morphology of the canyon [42,44,45].



Despite the results observed by Zhang et al. [32], who observed that urban morphology has a higher impact on the PET during the daytime rather than the nighttime, our study shows that the urban morphology accounts for the influence of the surrounding buildings' shade on the PET in all the seasons and at all times of the day. The urban morphology is also responsible for the sky view factor, which influences the local solar radiation and air temperature [46,47].

On the one hand, at night, a limited SVF results in heat storage due to the longwave radiation trapping, increasing the PET. This heat storage, which increases the air temperatures in urban areas when compared to the surrounding urban areas, is known as Urban Heat Island (UHI) [47]. Consequently, a high SVF is important to reduce the nighttime UHI as it was observed especially in Marstallplatz, which has lower PET values during the hot day nighttime. On the other hand, a high SVF provided by the large open area in the center of the Hohenzollernplatz is responsible for the high PET values at critical times of the hot day. The lower SVF of the Alpenplatz increases the nighttime PET, while decreasing the daytime PET values, an effect optimized by the presence of grass, trees, and the shadow of the buildings. The buildings' shade is also responsible for the thermal comfort in highly paved squares to counterbalance the longwave radiation and sparse vegetation, as observed at Alter Hof and Marstallplatz.

Our results highlight the importance of considering the seasonal variability of climatic conditions in the square design phase, taking advantage of different pavement and vegetation types depending on the square size and boundaries.

Regarding trees species, Xu et al. [22] also observed that deciduous trees are the most interesting species for urban squares as they have the ability to intercept most of the solar radiation in the summer and allow a relative increase in the SVF in winter, providing more solar radiation exposure.

Through a regression analysis it was possible to observe that in temperate climates surface temperature is really relevant only on warm days. While a highly paved surface was beneficial for increasing the PET due to the effect of radiation on cold days, the same type of surface had a negative effect on hot days. The regression analysis also revealed that the air temperature is the variable that affects the PET the most, affecting on average 93% of the results, except on the cold and humid day, where the wind speed is the variable that affects the PET values the most. The simulation also reinforced the importance of trees for decreasing the PET due to their shading potential on hot days, as already observed in other studies [1,48–50].

Unal Cilek and Uslu [51], in their study for Adana city, also observed the importance of square boundaries as an important factor to consider during the design phase to develop better-adapted climate strategies in urban squares. Wind protection and long wave radiation are the reasons why small squares present higher PET values. In a seasonal approach, is important to observe that a square protected from the wind is beneficial for thermal comfort on cold days, but detrimental under hot conditions when the wind has a positive influence, reducing the PET.

However, the most favorable PET results were obtained when the different types and elements of urban greenery and the presence or absence of pavement were previously considered in order to achieve the planned outcome. For instance, the presence of grass combined with the shade of the trees and the urban morphology, as also observed by Unal Cilek and Uslu [51] in Adana city, was responsible for lower PET values observed at Alpenplatz on hot days. On the other hand, longwave radiation is the factor that positively affects the PET on the cold day, as also observed by Irmak et al. [26] in Erzurum.

The highest values for longwave radiation on the cold day were observed close to the building walls at Alpenplatz, demonstrating that the square's size and shape affect the human thermal comfort in different ways depending on the season.

*Limitations of the Methodological Approach*

Due to the method used to select the typical days, the cold day and the cold and humid day cannot be compared directly.

During the simulations, it was observed that, despite the reliable output values of the PET, the Biomet tool of the ENVI-met model does not consider the variation in the clothing thermal insulation index (Clo) in the calculation of the PET. The Clo represents the insulation of clothing, which varies with season.

## 5. Conclusions

To understand how human thermal comfort is affected by the different elements of squares during the year, this study investigated five urban squares in Munich. They were analyzed through fieldwork and micrometeorological simulations, supplemented by multivariate regressions to understand the impact of each variable on the PET index. The study revealed that urban morphology is the factor with the greatest impact on the PET in all the studied cases.

All the analyzed variables are affected by the urban morphology (air temperature, wind speed, and radiation) and influence the PET by increasing or decreasing the thermal stress depending on the day and the time of day. For instance, wind-blocking by buildings increases the PET by up to 4 °C on the cold day, while the cooling effect of wind flow improves the thermal comfort on hot and warm days, emphasizing the importance of considering the impact of each design element in different seasons.

This study contributes to urban planning by illustrating the need to consider all the surrounding elements of a square and their dimensions to define the characteristics of new urban squares. To achieve human thermal comfort across the seasons in temperate climates, it may be necessary to combine conflicting solutions, such as the observed benefits of a building's shade in maximizing the radiation protection on hot days or their wind protection on cold days. Additionally, highlighting the importance of considering the height of buildings in high-density areas to ensure adequate daylight availability to open urban areas all year round and adequate SVF to avoid increasing the UHI in these areas particularly in the summer.

Future research should investigate the optimal green configuration to increase human thermal comfort in Munich's urban squares in different seasons. Further studies should also investigate how the insertion of typical greening designs and their microclimatic results, in different seasons, can contribute to increasing the human thermal comfort under changing climate conditions.

**Author Contributions:** Conceptualization, P.W.S.d.S. and S.P.; methodology, P.W.S.d.S.; software, P.W.S.d.S.; resources, P.W.S.d.S.; data curation, P.W.S.d.S.; writing—original draft preparation, P.W.S.d.S.; writing—review and editing, S.P. and D.D.; supervision, S.P. and D.D.; funding acquisition, S.P. All authors have read and agreed to the published version of the manuscript.

**Funding:** This study was realized within the project "Services of Urban Green at Public Sites in Munich" (TEW01C02P-75382) as part of the research center "Centre for Climate Ecology and Climate Adaptation (ZSK)" and funded by Bavarian State Ministry for the Environment and Consumer Protection.

**Data Availability Statement:** Not applicable.

**Acknowledgments:** The authors thank the Bavarian surveying authority for kindly geodatabase providing and Sabrina Erlwein for her kind help during the initial work discussion, mobile weather station assembly, and software support.

**Conflicts of Interest:** The authors declare no conflict of interest. The funders had no role in the design of the study; in the collection, analyses, or interpretation of data; in the writing of the manuscript; or in the decision to publish the results.

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
