# Peer review of "The Role of the Design of Public Squares and Vegetation Composition on Human Thermal Comfort in Different Seasons a Quantitative Assessment"

_land, doi:10.3390/land12020427_

Round 1

Reviewer 1 Report

Title: The role of the design of squares and vegetation composition on human thermal comfort in different seasons

Scientific research deals with the "unsolvable" problem at this latitude, hot summers and cold winters. The authors correctly designed the experiment, taking into account the diversity of city squares. Future studies or simulation should take into account the different distribution of vegetation in relation to the type of pavement and the surroundings. The authors drew the correct conclusions and include them in the paper.

Editorial errors

Standardize the time record to 12 hours - Figure 2 and 10

Add titles for the X axis - Figure 2, 10 and the Y axis - Figure 10.

Author Response

Dear Reviewer,
thank you for your helpful feedback.
The time record and axis in figures 2 and 10 (now 3 and 11) were done.
Please feel free to request anything else you consider necessary to improve the quality of our work.

Reviewer 2 Report

Overall, this study is based on good science and aims to answer some interesting and important research questions.  This manuscript is very interesting for publication. I have some recommendations that should be corrected before publication. 

1. Abstract_ all showed common knowledge, some interesting and quantify results should be concluded. There is still a lack of brief significance and conclusions. 

2. Introduction_ The literature reviews were not updated, many related studied were not concerned. A comprehensive literature review should be added to clearly reflect: 1) what the relevant research progress is and, 2) why your proposal is important. 

3. Methods_ A flow chart to describe the procedure of the experiment is needed. 

4. The discussion in this paper is incomplete. Discussion is an extension of the research results. Discussions should be based on the scientific nature and rationality of the results, combined with literature for in-depth analysis, and in-depth analysis of the mechanism of the results, the similarities and differences between the analysis and previous results, and attention should be paid to whether the comparison is consistent with similar reports. And explain why the result occurs and what the result means. There is a distinct lack of discussion regarding previous literature's results. Therefore, the authors need more comparisons with previous studies. 

5. The precise quantitative results of previous literature need to be mentioned a bit more so that readers can know if your research is consistent with previous studies. Finding studies that measured your results, in the same way, would also help to see differences between your research and others. In addition, I suggest you emphasize the contribution of research.

6.Conclusions_ Conclusions were all common knowledge. How to apply the results to real the diverse vegetation arrangements combined with the urban morphology characteristics? How to optimize human thermal comfort?Some prospective statements should be highlighted. 

7. There are still some problems with language expression in the manuscript, and the author is suggested to revise the language style further.

I encourage you to revise your work so that it has a chance to be published as I find your proposal interesting.

Best regards.

Author Response

Dear Reviewer,
thank you for your helpful feedback.
We made a general review of our paper, considering your relevant comments. Below are our comments:
Abstract: had a major language review and also a rearrangement of the sentences to better clarity.
Introduction: the literature review was updated and references 16,24,25,32,41,43,47,50 and 51 were added. 
Methods: the suggested flowchart was added.
Discussion: More comparisons with existing studies were included.
Conclusion: New inputs were added to answer your questions: “How to apply the results to real diverse vegetation arrangements combined with the urban morphology characteristics? How to optimize human thermal comfort? Some prospective statements should be highlighted.”
A language review by a native speaker was made in the full document.
Please feel free to request anything else you consider necessary to improve the quality of our work.

Reviewer 3 Report

The world is faced with a global climate emergency and extreme heat waves which can be exacerbated by urban design and morphology. So urban heat mitigation and outdoor thermal comfort improvement are definitely worth examining. Microclimatic conditions in urban environments is also of importance regarding human well-being and it is well documented that this problem can be intensified by UHI.

When it comes to "optimization" of UHI mitigation, and thermal comfort improvement, the article can significantly contribute to the related field as few studies have considered maximizing. However, the optimization is not adequately addressed and researched. The reason might be that the researchers have decided to cover a relatively broad topic investigating effects of various factors (including size, side walk type, and number of trees) on UHI, and also thermal comfort.

If the authors provide a discussion about results of this study in relation to that of previous articles and offer some new and original points, this manuscript can be a lasting contribution to the field. It is clear that the research lacks such a part, but there is room for improvement in this regard. The novelty is the main concern which can be addressed if results of this research can provide some new finding when compared with previous articles.

The abstract may be a useful summary, but the language and organization quality and should be improved. In addition, statements similar to the first line can be seen in some articles, it is better to start the abstract in a more interesting and accurate way.

Although there are relevant information in the introduction section, using subsections, paragraphing, tables, and graphs can help authors organize and summarize this section and avoid repetition. The article should mention the ideas in a way that the information supports the selection of simulation scenarios.
The article uses methods that are common and proven in this field, but the need for the design phase is not sufficiently justified. Though expansion of alterations in this phase of research can add a relatively new aspect to this article compared to previous ones, valuable information will not be lost even if this stage is omitted.

The conclusions are supported by the results. However, some results are repeated in this section without providing any priorities and practical guidelines on solutions to reduce UHI and improve thermal comfort. The explanation of the UHI effect, its causes and results, and human thermal comfort can be significantly organized. The idea about different SVF values and the number of trees is not clear. Rewrite these sentences and elaborate the idea. It is also unclear how the findings could be used in other urban spaces.

Both language and organization quality should be improved. There are errors related to word choice, coherence, progression, grammar, punctuation and paragraphing throughout the manuscript. So it is obvious that a proof reading should be conducted by a native English speaker. 

Author Response

Dear Reviewer,

thank you for your helpful feedback.

We made a general review of our paper, considering your relevant comments.

We are not sure if we understood completely your requests, and if our changes meet your expectations.

In any case, below are our main changes:

Abstract: had a major language review and also a rearrangement of the sentences to better clarity.

Introduction: the literature review was updated and references 16,24,25,32,41,43,47,50 and 51 were added. Also more comments to support our work were included.

Subtopics in the introduction section were not added, we tried to structure the section to achieve an approach that explores from a general aspect the specific topic of our research as follows:

-the importance of green spaces to improve human health.

-risks of heat stress in human health, and the importance of green to reduce heat stress.

-studies that consider how the greenery affects human thermal comfort (most commonly related to larger areas such as parks)

-studies that consider how squares influence the human thermal comfort

-most common studies are for summer condition

-studies that analyze seasonal conditions in other climates

-no studies found regarding the seasonal analysis of greenery in squares in a temperate climate.

The main focus was also clarified

Methods: a flowchart was added, to clarify the method.

And also the subtopics were reorganized as suggested.

Results: the definition of SVF was added (line 261)

Discussion: More comparisons with existing studies were included to reinforce our findings.

             The definition of UHI was provided (line 411)

Conclusion: Some prospective statements were added

A language review by a native speaker was made in the full document.

Please feel free to request anything else you consider necessary to improve the quality of our work.

Reviewer 4 Report

The article is interesting, suitable for Land magazine. The paper is well prepared, well written, and easy to read. I recommend publishing the post.

line 126 - start the table title with a capital letter ...considered variables... Considered variables...

Author Response

Dear Reviewer,
thank you for your helpful feedback.
Line 126 (now 134) was corrected.
Please feel free to request anything else you consider necessary to improve the quality of our work.

Round 2

Reviewer 2 Report

There are still some minor issues to be noted before publishing.

a. The indicator letter (a-g) in Figure 1 does not entirely coincide with the picture and needs to be modified. The letters in the picture should be in the upper left corner of the picture. The gray box under the letter should not exceed the picture.

b. The format of Table 2 needs to be adjusted. The table name does not need to be left blank with the table.

c. The indicator letter in Figure 6 does not entirely coincide with the picture and needs to be modified. The letters in the picture should be in the upper left corner of the picture. The gray box under the letter should not exceed the picture.

d. Figures 8, 9, 10, 17, and 18 have the insufficient resolution. The small text in the picture cannot be seen clearly.

The manuscript's content has been dramatically revised, but some details still need attention. Please take any point in the manuscript seriously. If your article is accepted and published, it will be seen by any scholar. Please ensure the high quality of the article.

Author Response

Dear reviewer,
Thank you for your improvement suggestions.
All the suggested images have improved quality resolution, and the introduction section had some paragraphs rewritten to improve clarity.
Kind regards

Reviewer 3 Report

The only observation that I would like to make concerns the structure of some sentences in the new part of the introduction: the sentences from "These findings suggest that urban morphology is the factor with the greatest impact on the human thermal comfort in urban squares, as also observed ...." should be rephrased to improve the structure of the text and make concepts clearer.

Also, the discussion and quality of figures 17 and 18 are very poor. The author should discuss these figures briefly and improve the image quality. 

Author Response

(The authors gave the same response as above.)
